health and disease and epidemiology/biomedical engineering/fluid mechanics

COVID-19, face covering, surgical mask, handmade mask, social distancing, respiratory droplets

**Authors for correspondence:**
Paul Digard
e-mail: paul.digard@roslin.ed.ac.uk
Ignazio Maria Viola
e-mail: i.m.viola@ed.ac.uk

†These authors contributed equally to this work.

# Face coverings and respiratory tract droplet dispersion

Lucia Bandiera[†,1], Geethanjali Pavar[1,†],
Gabriele Pisetta[1,†], Shuji Otomo[1,†], Enzo Mangano[1],
Jonathan R. Seckl[2], Paul Digard[3], Emanuela Molinari[4],
Filippo Menolascina[1] and Ignazio Maria Viola[1]

[1]School of Engineering, [2]Queen's Medical Research Institute, [3]The Roslin Institute, and [4]School of Informatics, University of Edinburgh, The King's Buildings, Edinburgh EH9 3FB, UK

 SO, 0000-0002-0344-3961; IMV, 0000-0002-3831-8423

Respiratory droplets are the primary transmission route for SARS-CoV-2, a principle which drives social distancing guidelines. Evidence suggests that virus transmission can be reduced by face coverings, but robust evidence for how mask usage might affect safe distancing parameters is lacking. Accordingly, we set out to quantify the effects of face coverings on respiratory tract droplet deposition. We tested an anatomically realistic manikin head which ejected fluorescent droplets of water and human volunteers, in speaking and coughing conditions without a face covering, or with a surgical mask or a single-layer cotton face covering. We quantified the number of droplets in flight using laser sheet illumination and UV-light for those that had landed at table height at up to 2 m. For human volunteers, expiratory droplets were caught on a microscope slide 5 cm from the mouth. Whether manikin or human, wearing a face covering decreased the number of projected droplets by less than 1000-fold. We estimated that a person standing 2 m from someone coughing without a mask is exposed to over 10 000 times more respiratory droplets than from someone standing 0.5 m away wearing a basic single-layer mask. Our results indicate that face coverings show consistent efficacy at blocking respiratory droplets and thus provide an opportunity to moderate social distancing policies. However, the methodologies we employed mostly detect larger (non-aerosol) sized droplets. If the aerosol transmission is later determined to be a significant driver of infection, then our findings may overestimate the effectiveness of face coverings.

# 1. Introduction

SARS-CoV-2 is primarily transmitted from virus-laden fluid droplets ejected from the mouth of an infected carrier. These droplets are either inhaled by a recipient, deposited on the recipient's mouth or conjunctiva, or deposited on a surface (thereby generating a fomite) and then mechanically transmitted through physical contact [1]. Droplet diameter varies according to the mode of exhalation (e.g. quiet breathing versus coughing) but ranges from approximately 100 nm to 1 mm [2,3]. Intuitively, larger droplets potentially contain more virus, but the infectivity of viruses may vary between droplet size classes due to the influence of drying, making it extraordinarily difficult to determine the dominant mode of transmission [4]. Evidence indicates that transmission occurs both through the smallest droplets [5,6], known as an aerosol, as well as the direct deposition on the recipient's mouth, nose or conjunctiva of the largest droplets [7,8]. This paper focuses on the effect of face coverings in mitigating the dispersion of the largest droplets.

Recent reviews [9–11] suggest that face coverings are effective in decreasing the risk of infection. This has been inferred through epidemiological studies [12], tests with animals [13] and physical tests. For example, Viola *et al.* [14] showed that face coverings decreased the front throughflow of aerosol droplets by about an order of magnitude.

Comparatively, there is weaker evidence that face coverings mitigate the dispersion of large respiratory droplets. Rodriguez-Palacios *et al.* [15] sprayed bacterial-suspension droplets with a diameter from 20 to 900 µm through different household textiles and showed that this could mitigate large droplet dispersal. Yet, to allow an evidence-based assessment of what the appropriate social distance is when masks are worn, face-fitted masks must be tested with realistic respiratory airflow jets and with human volunteers.

Recently, Anfinrud *et al.* [16] demonstrated that large respiratory droplets ejected by a person speaking can be visualized by laser sheet illumination. Successively Fischer *et al.* [17] used this technique to compare the effectiveness of different face masks in filtering respiratory droplets. A person spoke in front of a hole in an enclosed box, where droplets passing through a laser sheet were imaged and counted for different face coverings. These experiments demonstrated that face coverings can be effective in mitigating the dispersion of droplets ejected by a person speaking. On the other hand, these studies did not provide information on the distance travelled by the droplets, and thus on how face coverings can be used to review social distancing guidelines.

We adopted a similar technique to quantify the in-flight droplets ejected by an anatomically realistic manikin, with and without face covering, at several distances up to 2 m from the source. We considered both speaking and coughing conditions. Furthermore, we complemented these measurements with two other independent measuring techniques, UV light imaging and microscopy, which resulted in consistent results. Finally, we measured with microscopy the droplets ejected by six individuals and the results corroborated those of the manikin.

# 2. Methods

## 2.1. Speech and cough simulator

The same simulator used by Viola *et al.* [14] was used. Airflow was generated using an air compressor capable of delivering up to $180 \, l \, min^{-1}$, while a 100 µM fluorescein solution in water for droplet generation was supplied by a TCS M400S micropump with a variable output range of $0–2.7 \, l \, min^{-1}$. Liquid and airflows were connected to a purpose-built droplet generation system fitted inside an anatomically realistic, adult, medical simulation manikin's mouth (Resusci Anne QCPR) that creates controllable droplet sizes. Using shadow imaging (electronic supplementary material), we estimated that the manikin ejected a continuous distribution of droplets ranging from less than $26 \pm 7$ µm, which is the size of the smallest droplet we could measure and that remained airborne, and up to $590 \pm 7$ µm. There is no agreement in the literature on the average droplet size distribution of humans while speaking and coughing [18], but several measurements suggest that the median is of the order of 10 µm, and droplets of 500 µm occur at a frequency that is one or two orders of magnitude lower [19]. Hence the manikin's range includes the largest aerosol droplets and all droplets with ballistic trajectories ejected by humans.

Air/liquid flows were ejected from a 2 cm diameter circular 'tracheal' opening with a velocity comparable to those of a person either speaking [20] ($1 \, m \, s^{-1}$) or coughing [19] ($10 \, m \, s^{-1}$). To ensure a

significant number of landed particles could be detected prior to their evaporation, droplets were ejected from the manikin at a higher volume rate compared to a human being. Specifically, 20 min of speech test in the manikin corresponded to a subject counting from 1 to more than 10 000; 10 min of coughing tests modelled 60 000 human coughs.

Masks were either surgical or single-layer woven cotton (see electronic supplementary material). Examination of samples by phase-contrast microscopy showed that the cotton masks had a close weave with gaps of around 50 µm, while the multiple layers of less densely woven material in the surgical mask left a broader spectrum of pore sizes (electronic supplementary material, figure S1).

## 2.2. Laser imaging of droplets in flight

We projected a thin laser sheet along the vertical plane through the mouth of the manikin and used a photographic camera to capture the light scattered by droplets passing within this plane. A 2.5 W diode-pumped continuous wave laser (532 nm) running at 40% of maximum power illuminated a plane perpendicular to the floor along the airflow jet axis. An 8-bit CCD camera with a resolution of $2056 \times 2060$ pixels with a Nikon 50 mm f/2 lens was used to image a physical plane of $137.5 \times 137.8$ mm, with a resolution of 67 µm pixel$^{-1}$. A 60 ms exposure time was used to count particles under all conditions except coughing without a mask, where 30 ms was used. A total of 100 images from six replicate experiments were used for particle counting analysis. The light scattered by fluorescein made droplets appear larger than their actual size. Using a shadow imaging technique, we verified that the laser imaging visualized all droplets large enough to fall ballistically (see electronic supplementary material).

## 2.3. UV-light imaging

Paper sheets were placed on a table 0.426 m below the manikin to cover areas of $0.84 \times 0.6$ m for speaking and $2.1 \times 0.3$ m for coughing. Speaking and coughing tests without a face covering lasted 2 and 1 min, respectively, while durations of 20 and 10 min were used when the manikin was wearing a handmade or surgical mask. At least six replicates were run for each test (speaking, coughing) and condition (presence/absence of face covering). After each test, paper samples were placed on a Safe Imager 2.0 Blue-Light Transilluminator (InvitrogenTM G6600) to visualize deposited droplets. Images were acquired using an iPhone 7 camera (f/1.8, $4032 \times 3024$ pixels) with a resolution of 65 µm pixel$^{-1}$. Images were imported into Inkscape to reconstruct the continuous sample and to equalize the pictures across the whole sample. Upon binarization of the reconstructed image with a manually defined threshold, droplets were counted in each grid element using Fiji's edge-detection algorithm.

## 2.4. Microscopy

Samples caught on glass microscope slides were imaged with a CFI Plan Fluor $20 \times$ objective mounted on a Nikon Eclipse Ti inverted microscope, equipped with an iXon Ultra 888 EMCCD camera, resulting in a resolution upon magnification of 0.65 µm pixel$^{-1}$. For each sample, 210 fields of view, covering an area of 0.85 cm$^2$ in total, were acquired in the FITC channel (gain 70, exposure time 100 ms) using NIS-Elements. Samples collected in tests involving human subjects were imaged with an Andor Zyla sCMOS camera and the same objective and microscope as above, resulting in a resolution upon magnification of 0.33 µm pixel$^{-1}$. We acquired 153 fields of view, corresponding to 0.62 cm$^2$. All images were downscaled by a factor of five prior to their analysis in scikit-image, Python. Droplets in each sample were counted as the markers of the watershed algorithm, which allows segmentation of touching objects in an image. A Gaussian smoothing filter was applied (standard deviation of five) to reduce pixel noise before a thresholding procedure using a manually selected value. Following morphological opening with a Boolean kernel of size seven, droplet centres were identified as the local maxima of the distance transform of the image, with the stipulation that centres should be more than 15 pixels from each other.

## 2.5. Tests with human volunteers

Six volunteers performed two rounds of coughing (1 min each) and speaking (3 min each) tasks, with and without a surgical mask. Speech tasks were performed reading a provided sample text, to ensure results were not biased by the personal choice of words. On the 2 days of the trial experiment, the

order in which the tasks were performed was reversed. This research protocol was approved by the University of Edinburgh's Human Research Ethical Review Committee and all human participants gave written informed consent.

## 2.6. Statistical analysis

The mean and the standard error of the mean (s.e.m.) of the data are presented. Due to the inhomogeneity of variance, results were assessed by Bonferroni-corrected Kolmogorov–Smirnov statistical tests applied to pairwise combinations of no-mask/handmade mask, no-mask/surgical mask and handmade mask/surgical mask. Individual statistical tests were performed for each distance from the source.

# 3. Results

## 3.1. Imaging droplets in flight

We recorded images at eight different positions, whose vertical and horizontal distances are presented in figure 1a. Fluorescent droplets appeared as segments on the images, with a length proportional to their speed (electronic supplementary material, figure S2). We observed three types of droplets: droplets smaller than approximately 30 μm that remained airborne following air currents in the room, droplets larger than approximately 170 μm that fell ballistically with trajectories similar to the red lines in figure 1a, and intermediate size droplets which could show either of the two behaviours or fall with a non-ballistic trajectory. The size of the droplets was assessed using shadow imaging technique (electronic supplementary material).

Visually, the impact of placing a mask on the manikin was obvious, especially under coughing conditions (figure 1b). To quantify the data, we counted the number of particles crossing the lower edge of the field of view at positions 1 to 7, therefore extrapolated to have deposited on the table. Data were compiled from 100 images taken from six replicate runs, each with no face covering, with a surgical mask or with a single-layer face covering. We used a new mask for each repeat to account for potential variability in the masks. The absolute particle count was divided by the length of the lower edge of the field of view and by the total period of observation to provide the rate of deposition of particles along the centreline of the table. Tests modelling speaking without a mask showed an exponential decline in particle deposition rate with distance, with none reaching the 2 m mark (figure 1c), while the distribution peaked at 1 m and dropped more slowly for coughing (figure 1d). When the manikin was fitted with either surgical or handmade masks, not one droplet was found to fall through the lower edge of the fields of view in either speaking or coughing simulations. Therefore, both face masks were largely impermeable to droplets larger than 20–30 μm.

## 3.2. Droplet deposition

To provide an overview of the spread of droplets over the table, we first used UV light to image the distribution of fluorescent particles that accumulated on white paper placed in front of the manikin. An example result for a speaking test is shown in electronic supplementary material, figure S3. We measured droplet deposition rate according to distance, presented as the number of particles divided by the sampling area and the duration of the experiment by counting the droplets every 5 and 15 cm from the manikin for speaking and coughing, respectively. Each experiment was performed six times with and without both types of face covering, using a new mask for each repeat. The numbers of surface-deposited droplets were consistent with the measurements of falling droplets (figure 2a,b). Once again, the presence of a face covering very effectively blocked droplet deposition. However, because we measured droplets on a larger area and longer duration (up to 20 min) than the flight tests, we could capture less frequent events. With both mask types, small numbers of droplets (up to seven) were seen at various distances from the manikin, but these values were four and five orders of magnitude lower than those observed without a mask for speaking and coughing, respectively. Hence, these represent events with a frequency of one in 10 000 and one in 100 000, respectively.

To further corroborate our results, we imaged deposited droplets using microscopy. To this end, the manikin was set to emulate speaking or coughing as before and fluid droplets were collected on glass slides positioned at varying distances (0.25, 0.5 and 0.75 m for speaking conditions and 0.5, 1, 1.5 and

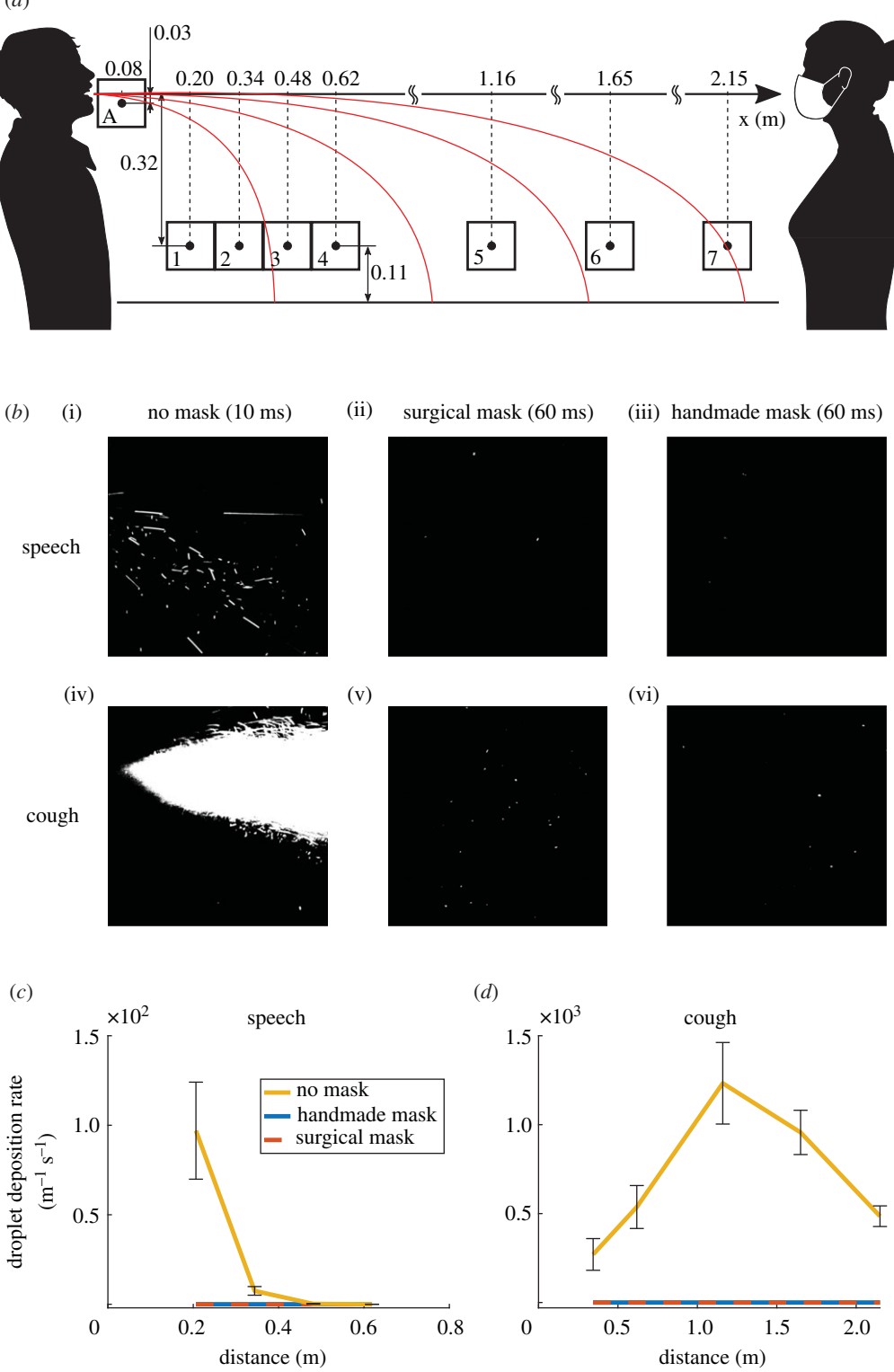

**Figure 1.** Laser imaging of respiratory droplets in flight. (*a*) Schematic diagram of the experimental set-up. Boxes indicate imaging windows. (*b*) Examples of images captured at position A (directly in front of the mouth) for speaking (i, ii, iii) and coughing (iv, v, vi), without mask (i, iv), with the surgical mask (ii, v) and with the handmade mask (iii, vi). (*c,d*) Droplet deposition rate over the table centreline versus the horizontal distance from the manikin's mouth in (*c*) speaking and (*d*) coughing conditions. Data were estimated from the count of imaged droplets crossing the lower edge of the field of view per unit time and are reported as the mean ±1 s.e.m. of six independent replicates. (*c*) Both mask types statistically significantly reduced the droplet deposition rate at 0.21 m from the mouth of the source ( $p = 0.016$ in both cases). Past 0.21 m, too few droplets were detected also without mask to observe statistically significant differences among the tested conditions. (*d*) No significant differences were detected between mask types, but both mask types statistically significantly reduced the droplet deposition rate at every tested position ( $p = 0.019$ ).

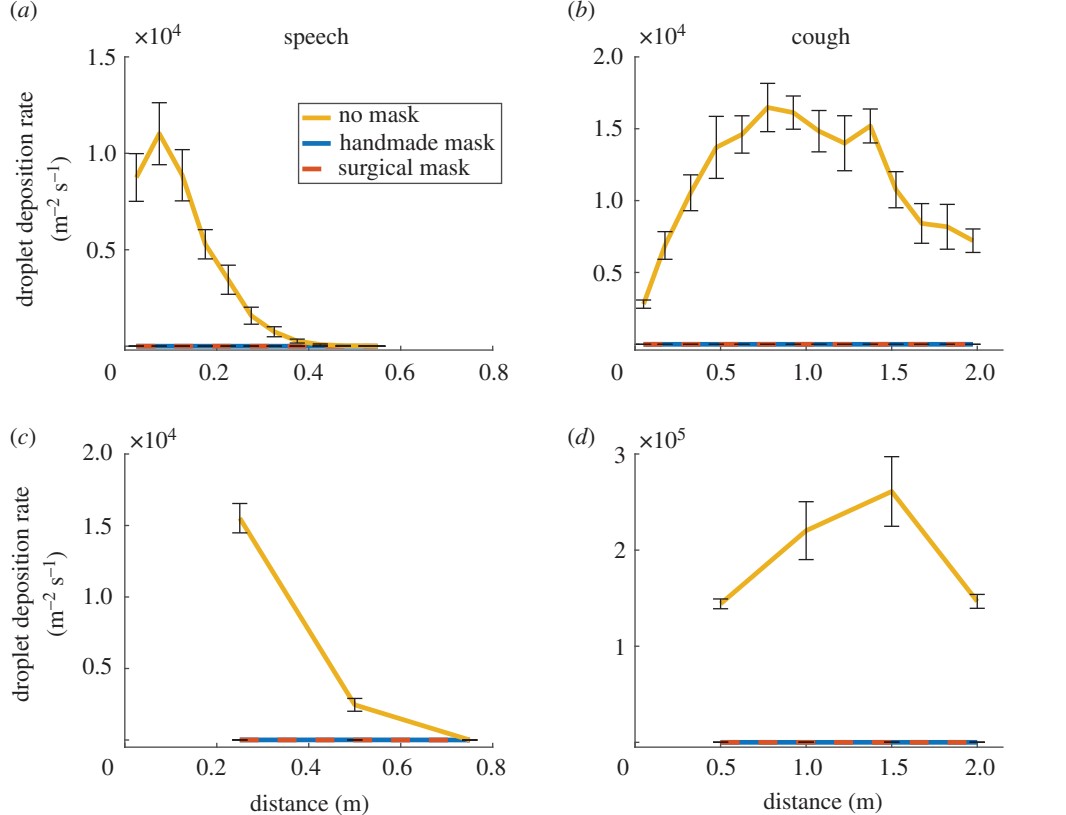

**Figure 2.** Surface deposition of droplets. Droplet deposition rate versus horizontal distance from the manikin's mouth in (*a,c*) speaking or (*b,d*) coughing conditions, measured by (*a,b*) UV illumination of paper sheets or (*c,d*) microscopic imaging of glass slides placed on the table centre line. Data are the mean ±1 s.e.m. of six independent repeats. Statistical analyses indicated that for (*a*), both mask types significantly reduced the droplet deposition rate at up to 47.5 cm from the source (*p* = 0.047). For (*b*), droplet deposition rate with and without masks were not significantly different (*p* = 0.055). For (*c*), both mask types significantly reduced droplet deposition rate up to 0.5 m from the source (*p* = 0.012). Past that point, droplets were undetectable in all cases. For (*d*), both mask types significantly reduced droplet deposition rate at all distances from the source (*p* = 0.016). No statistically significant difference was observed between the two mask types in any test.

2 m for coughing conditions) from the manikin. We repeated each experiment six times with and without masks, using a new mask for each test. After each run, the glass slides were collected and imaged with a 20 × objective to count the number of droplets deposited in both cases. We then reconstructed the spatial distribution of droplets with and without masks (figure 2*c,d*). Consistent with other means of measurement, both types of face covering were extremely effective at reducing the particle deposition rate. As with UV imaging of droplets deposited on white paper, this technique also allowed a long experimental duration (20 min) and thus the capture of statistically rare events. With both mask types, we observed a few (up to four) droplets at various distances from the manikin, including at 2 m for coughing. However, this number was again 4–5 orders of magnitude lower than the number of droplets observed without a mask for speaking and coughing, respectively.

## 3.3. Droplet deposition from human volunteers

Results obtained from a manikin, one might argue, are only indicative due to the wide variability in the expiration parameters for speech and coughing in humans [18,21]. Accordingly, we performed experiments using four male and two female volunteers (aged between 30 and 45) to either read a sample script for 3 min or cough for 1 min. For each test, a glass slide was placed vertically, 5 cm in front of the individual's mouth. At the end of each experiment, the sample was imaged using widefield microscopy to count the droplets generated by the subject. Each speaking and coughing test was performed twice with and without a surgical mask on separate days, reversing the order of the tests on day two. In contrast with the manikin tests, there was a large variability in the number of

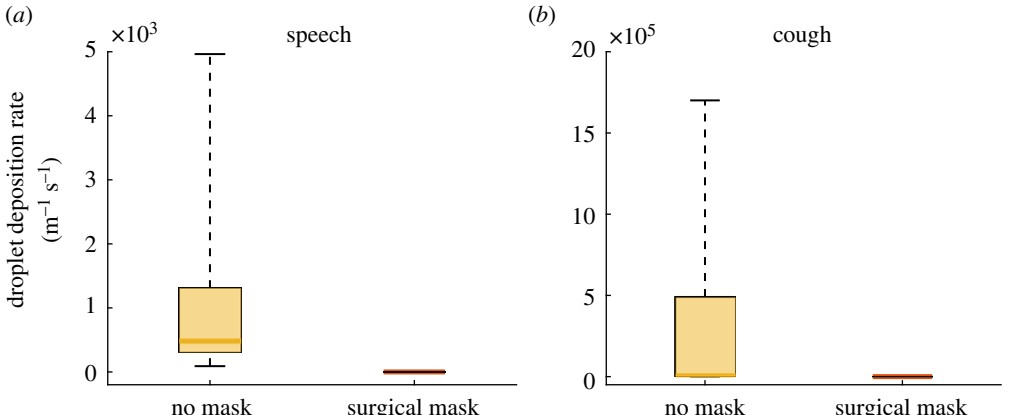

**Figure 3.** Droplet deposition from human volunteers. Droplet deposition rate in human volunteers for the no-mask and surgical mask cases under (*a*) speech and (*b*) coughing conditions. Data represent the number of droplets imaged in widefield microscopy on a glass slides placed vertically at a distance of 5 cm from the human volunteer performing the task. To estimate the deposition rate, droplet counts were divided by the imaged area and duration of the speech or coughing task. Measurements are plotted as a box between the 25th and 75th percentiles with a mark indicating the median. The whiskers extend over the full data range. Statistical tests showed $p = 2.3 \times 10^{-6}$ for both cases, thus rejecting the null hypothesis that the two samples were drawn from the same distribution. Therefore, the use of surgical masks statistically significantly reduced droplet deposition rate.

droplets emitted by the individual subjects in the absence of a mask (figure 3). However, for all subjects, we did not find a single droplet when a mask was worn.

## 4. Discussion

We employed three independent, quantitative techniques to measure the droplet deposition rate at increasing distance from a manikin. Without any face covering, we measured distributions of deposition rates comparable to those reported in the literature [22–24]. When the manikin wore either of the two face masks, we observed that less than one in 1000 particles escaped for both speaking and coughing. To further demonstrate that our results could be applied to real people speaking and coughing, we tested six individuals using microscopy. Without masks, we measured between tens and thousands of particles for speaking and coughing without a mask, respectively. Conversely, we found zero particles with the surgical mask, both for speaking and coughing.

Overall, these data demonstrate that face masks are highly effective at reducing exhalation of large respiratory droplets. As these droplets are likely to be the main driver of SARS-CoV-2 transmission [7,8,12], our data suggest that the wearing of masks can substantially reduce the probability of an infected person transmitting the virus. In this study, however, we focused only on the large respiratory droplets that land on a surface within a few seconds. According to the shadow imaging estimation of droplet size applied to the laser acquisition (electronic supplementary material, S4), these correspond to particles with diameters between $170 \pm 7$ and $590 \pm 7$ µm. Droplets as small as $26 \pm 7$ µm were detected with the laser but excluded from the analysis due to their non-ballistic trajectory. Other studies such as Fischer *et al*. [17], which focuses on in-flight droplets that include smaller droplets and aerosol, are likely to show a lower level of effectiveness depending on the fabric type. The aerosol is now widely accepted as a contributory route of SARS-CoV-2 virus transmission [5], and if this is later determined to be the main driver of infection, then our findings may overestimate the effectiveness of face coverings.

Air was ejected at velocities and flow rates within the range of those observed from real individuals [19,20] but droplets were ejected at a higher volume rate. Hence, the physical interaction between the droplets and the airflow jet [21,24] was not correctly represented. However, this was necessary to ensure a high volume of particles and thus robust statistics, in a sufficiently short time frame to minimize evaporation of landed droplets and contamination with other particles in the environment. By counting from 1 to 100, a person ejects between 10 and 100 mg of water [23], while our manikin ejected about 10 mg s$^{-1}$. Thus, each second of test corresponded to a person counting at least from 1 to 100. The duration of the tests was as high as 20 min; hence we modelled a person counting up from 1 to more than 10 000. In one cough, a person ejects about 1 mg of fluid [23], while our manikin

ejected $100\,\text{mg s}^{-1}$. We tested a cough for as long as 10 min, which is equivalent to 60 000 coughs. Furthermore, it was not possible to test if droplets escaped the mask and landed outside of the table. Viola *et al*. [14] showed that surgical and handmade masks can lead to lateral and backwards air jets. If a large droplet was carried by these jets, we would have not detected it. Finally, we tested simple single-layer handmade masks, but it should be recognized that there is a wide range of handmade masks and some might not be fit for purpose.

In conclusion, these experiments demonstrate that both surgical and simple handmade masks such as a single-layer cotton mask can suppress the risk of direct person-to-person virus transmission through large droplet deposition. The data do not allow us to draw conclusions on the risks of virus transmission through aerosol inhalation. Assuming that SARS-CoV-2 virus transmission through aerosol is small compared to through large droplets, these results suggest that physical distancing can be reduced with the use of face coverings.

Ethics. Research Ethics and Integrity Assessment approved by the Director of Research of the School of Engineering, Prof. Bernie Mulgrew, The University of Edinburgh.

Data accessibility. Processed and raw data is available on Edinburgh DataShare: https://datashare.is.ed.ac.uk/handle/10283/3729.

Authors' contributions. L.B., F.M., S.O., Ge.P. and Ga.P. undertook the experiments. E.M. led the design and manufacturing of the speech and cough simulator. F.M. conceived and led the microscopy investigation. P.D., F.M., E.M., J.R.S. and I.M.V. designed the experiments. I.M.V. coordinated the project. E.M. wrote the first draft of this document that was edited, reviewed and approved by all the authors.

Competing interests. We declare we have no competing interests.

Funding. L.B. is supported by the UK Engineering and Physical Sciences Research Council (EPSRC), grant no. EP/P017134/1. Ge.P. and Ga.P. are supported by the EPSRC grant EP/L016680/1, while E.M. by the EPSRC grant no. EP/S02431X/1. S.O.'s scholarship is funded by the Japan Student Services Organization. F.M. is supported by the European Commission (766840) and the EPSRC (EP/S001921/1 and EP/R035350/1), while P.D. is funded by the UK Biotechnology and Biological Sciences Research Council (BB/P013740/1).

Acknowledgements. The authors are grateful to Dr B. Peterson (School of Engineering) and Dr A. Nila (LaVision) for their advice on the laser tests and for lending related equipment, and to Dr K. Dunn (School of Engineering), Prof. F. Denison (Queen's Medical Research Institute) and Dr F. Mehendale (Usher Institute) for procuring the tested face masks.

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
