## [Reviewer comments · Royal Society Open Science]

Review History

RSOS-201663.R0 (Original submission)

Review form: Reviewer 1 (Timothy Corcoran)

Is the manuscript scientifically sound in its present form?

Yes

Are the interpretations and conclusions justified by the results?

Yes

Is the language acceptable?

Yes

Do you have any ethical concerns with this paper?

No

Have you any concerns about statistical analyses in this paper?

No

Recommendation?

Major revision is needed (please make suggestions in comments)

Comments to the Author(s)

Intro: Viola et al. [10] showed that face coverings decreased the front throughflow of the aerosol jet by about an order of magnitude. "Aerosol jet" is confusing. Suggest referring to either aerosol droplets and particles or the air jet.

Intro: The introduction would benefit from a discussion of the size characteristics or respiratory aerosols and the viral content associated with different size classes.

Intro: If there is enough space the intro would benefit from a discussion how deposition characteristics vary with aerosol size across the size ranges relevant to respiratory aerosols (roughly 1mm – 1 μ m). Sedimentation times of a 1mm vs 1 μ m aerosol, etc.

Methods: A major issue with the paper is that the droplet size characteristics of the test aerosol emitted from the manakin are not well described or compared to aerosol characteristics of the respiratory aerosols from a person. This needs to be done in some form. It would seem that the authors have the data to make this comparison, but literature assessments of respiratory aerosol size distributions could be used if necessary.

Methods: The methods should at least briefly discuss the fact that higher liquid output rates are being used by the investigators in order to facilitate detectability. This is described thoroughly in the discussion, but it should be mentioned in the methods as well. Describe the exact number of coughs being simulated, etc. from the start.

Results: I'd suggest the authors consider some characterizations of droplet mass rather than just droplet number, since this is the relevant parameter in terms of transmitting virus, as larger droplets will carry significantly more virus. Mass based diameters are almost exclusively used in discussions of medicinal aerosols for similar reasons since larger aerosols will transport higher drug doses. What percentage of total aerosol mass is transported to different distances?

Results Fig 1: Would suggest clarifying in the caption that deposition is based on image estimates.

Results Fig 3: Caption should describe the method used to capture droplets

Discussion: The authors should discuss as specifically as possible the lower limit of aerosol size detection for their method as a limitation. This is well described in the supplement but some baseline information should be included in the discussion as well.

Would suggest in future studies investigators consider the lateral transmission of fine aerosols from the sides and top of the masks.

Review form: Reviewer 2

Is the manuscript scientifically sound in its present form?

Yes

Are the interpretations and conclusions justified by the results?

Yes

Is the language acceptable?

Yes

Do you have any ethical concerns with this paper?

No

Have you any concerns about statistical analyses in this paper?

No

Recommendation?

Accept with minor revision (please list in comments)

Comments to the Author(s)

Aerosol transmission is mostly accepted for COVID-19 now - the authors need to acknowledge this and update their literature review/Introduction/Discussion, to reflect this - but emphasise that they are focusing on the large droplet containment properties of various masks.

Also, I've reviewed this paper previously and found the video very unhelpful - the orientation is not very clear and the particles are not really visible.

They need to improve this or just delete it.

Decision letter (RSOS-201663.R0)

Dear Dr Viola

On behalf of the Editors, we are pleased to inform you that your Manuscript RSOS-201663 "Face Coverings and Respiratory Tract Droplet Dispersion" has been accepted for publication in Royal Society Open Science subject to mandatory revision in accordance with the referees' reports. Please find the referees' comments along with any feedback from the Editors below my signature.

Please submit your revised manuscript and required files (see below) no later than 7 days from today's (ie 25-Nov-2020) date. Note: the ScholarOne system will 'lock' if submission of the revision is attempted 7 or more days after the deadline. If you do not think you will be able to meet this deadline please contact the editorial office immediately.

on behalf of Dr Michael Doube (Associate Editor) and R. Kerry Rowe (Subject Editor)
openscience@royalsociety.org

Associate Editor Comments to Author (Dr Michael Doube):

Associate Editor: 1

Comments to the Author:

Two reviewers have seen the manuscript and are broadly supportive, with a number of comments that you should be able to address with changes to your text. You should ensure that sufficient information is included to allow readers to translate the findings from your simulation into the context of real respiratory aerosols, and update the text to include relevant context that has emerged since the manuscript was first submitted.

Reviewer comments to Author:

Reviewer: 1

Comments to the Author(s)

Intro: Viola et al. [10] showed that face coverings decreased the front throughflow of the aerosol jet by about an order of magnitude. "Aerosol jet" is confusing. Suggest referring to either aerosol droplets and particles or the air jet.

Intro: The introduction would benefit from a discussion of the size characteristics or respiratory aerosols and the viral content associated with different size classes.

Intro: If there is enough space the intro would benefit from a discussion how deposition characteristics vary with aerosol size across the size ranges relevant to respiratory aerosols (roughly 1mm - 1 μ m). Sedimentation times of a 1mm vs 1 μ m aerosol, etc.

Methods: A major issue with the paper is that the droplet size characteristics of the test aerosol emitted from the manikin are not well described or compared to aerosol characteristics of the respiratory aerosols from a person. This needs to be done in some form. It would seem that the authors have the data to make this comparison, but literature assessments of respiratory aerosol size distributions could be used if necessary.

Methods: The methods should at least briefly discuss the fact that higher liquid output rates are being used by the investigators in order to facilitate detectability. This is described thoroughly in the discussion, but it should be mentioned in the methods as well. Describe the exact number of coughs being simulated, etc. from the start.

Results: I'd suggest the authors consider some characterizations of droplet mass rather than just droplet number, since this is the relevant parameter in terms of transmitting virus, as larger droplets will carry significantly more virus. Mass based diameters are almost exclusively used in discussions of medicinal aerosols for similar reasons since larger aerosols will transport higher drug doses. What percentage of total aerosol mass is transported to different distances?

Results Fig 1: Would suggest clarifying in the caption that deposition is based on image estimates.

Results Fig 3: Caption should describe the method used to capture droplets

Discussion: The authors should discuss as specifically as possible the lower limit of aerosol size detection for their method as a limitation. This is well described in the supplement but some baseline information should be included in the discussion as well.

Would suggest in future studies investigators consider the lateral transmission of fine aerosols from the sides and top of the masks.

Reviewer: 2

Comments to the Author(s)

Aerosol transmission is mostly accepted for COVID-19 now - the authors need to acknowledge this and update their literature review/Introduction/Discussion, to reflect this - but emphasise that they are focusing on the large droplet containment properties of various masks.

Also, I've reviewed this paper previously and found the video very unhelpful - the orientation is not very clear and the particles are not really visible.

They need to improve this or just delete it.

===PREPARING YOUR MANUSCRIPT===

Your revised paper should include the changes requested by the referees and Editors of your manuscript. You should provide two versions of this manuscript and both versions must be provided in an editable format:
one version identifying all the changes that have been made (for instance, in coloured highlight, in bold text, or tracked changes);
a 'clean' version of the new manuscript that incorporates the changes made, but does not highlight them. This version will be used for typesetting.
Please ensure that any equations included in the paper are editable text and not embedded images.

===PREPARING YOUR REVISION IN SCHOLARONE===

To revise your manuscript, log into <https://mc.manuscriptcentral.com/rsos> and enter your Author Centre - this may be accessed by clicking on "Author" in the dark toolbar at the top of the

page (just below the journal name). You will find your manuscript listed under "Manuscripts with Decisions". Under "Actions", click on "Create a Revision".

<https://royalsociety.org/journals/authors/author-guidelines/#supplementary-material> to include a suitable title and informative caption. An example of appropriate titling and captioning may be found at https://figshare.com/articles/Table_S2_from_Is_there_a_trade-off_between_peak_performance_and_performance_breadth_across_temperatures_for_aerobic_sc_ope_in_teleost_fishes_/3843624.

Author's Response to Decision Letter for (RSOS-201663.R0)

See Appendix A.

Decision letter (RSOS-201663.R1)

Dear Dr Viola,

It is a pleasure to accept your manuscript entitled "Face Coverings and Respiratory Tract Droplet Dispersion" in its current form for publication in Royal Society Open Science.

COVID-19 rapid publication process:

We are taking steps to expedite the publication of research relevant to the pandemic. If you wish, you can opt to have your paper published as soon as it is ready, rather than waiting for it to be published the scheduled Wednesday.

This means your paper will not be included in the weekly media round-up which the Society sends to journalists ahead of publication. However, it will still appear in the COVID-19 Publishing Collection which journalists will be directed to each week (<https://royalsocietypublishing.org/topic/special-collections/novel-coronavirus-outbreak>).

If you wish to have your paper considered for immediate publication, or to discuss further, please notify openscience_proofs@royalsociety.org and press@royalsociety.org when you respond to this email.

Best regards,

on behalf of Dr Michael Doube (Associate Editor) and R. Kerry Rowe (Subject Editor)
openscience@royalsociety.org

Appendix A

A marked version of the manuscript is attached with edited text highlighted in yellow.

Reply to the Reviewer 1's comments

1. Intro: Viola et al. [10] showed that face coverings decreased the front throughflow of the aerosol jet by about an order of magnitude. "Aerosol jet" is confusing. Suggest referring to either aerosol droplets and particles or the air jet.

This is now corrected throughout the manuscript.

2. Intro: The introduction would benefit from a discussion of the size characteristics or respiratory aerosols and the viral content associated with different size classes.

This is now discussed in the first paragraph of the paper, as well as in the first paragraph of the Methods, where the droplet size distribution ejected by a manikin is compared with that for humans reported in the literature.

3. Intro: If there is enough space the intro would benefit from a discussion how deposition characteristics vary with aerosol size across the size ranges relevant to respiratory aerosols (roughly 1mm – 1 μ m). Sedimentation times of a 1mm vs 1 μ m aerosol, etc.

This is a very good suggestion. We considered how this could have been added but in the interest of brevity we decided not to include such a discussion so as not to dilute the focus of the paper.

4. Methods: A major issue with the paper is that the droplet size characteristics of the test aerosol emitted from the manikin are not well described or compared to aerosol characteristics of the respiratory aerosols from a person. This needs to be done in some form. It would seem that the authors have the data to make this comparison, but literature assessments of respiratory aerosol size distributions could be used if necessary.

In the Methods (first paragraph) we now specify the droplet size range measured with shadow imaging and we compare it with that reported in the literature for a person speaking and coughing.

5. Methods: The methods should at least briefly discuss the fact that higher liquid output rates are being used by the investigators in order to facilitate detectability. This is described thoroughly in the discussion, but it should be mentioned in the methods as well. Describe the exact number of coughs being simulated, etc. from the start.

This is now clarified in the second paragraph of the Methods.

6. Results: I'd suggest the authors consider some characterizations of droplet mass rather than just droplet number, since this is the relevant parameter in terms of transmitting virus, as larger droplets will carry significantly more virus. Mass based diameters are almost exclusively used in discussions of medicinal aerosols for similar reasons since larger aerosols will transport higher drug doses. What percentage of total aerosol mass is transported to different distances?

Intuitively, one might expect that larger droplets would carry more virus and thus carry a higher risk of infection. Unfortunately, there are many other variables and unknowns that prevent this

simple association. Droplet size may well vary according to where the liquid arises from in the respiratory tract, but virus replication from mouth through to alveoli and hence the seeding load is not uniform. Furthermore, infectivity will vary according to the composition of the droplet (again, subject to non-uniform variation) and drying time. We have included a new citation to a good review of this subject in the introduction, [4].

7. Results Fig 1: Would suggest clarifying in the caption that deposition is based on image estimates.

Done.

8. Results Fig 3: Caption should describe the method used to capture droplets

Done.

9. Discussion: The authors should discuss as specifically as possible the lower limit of aerosol size detection for their method as a limitation. This is well described in the supplement but some baseline information should be included in the discussion as well.

This is now clarified in the Discussion (second Paragraph).

10. Would suggest in future studies investigators consider the lateral transmission of fine aerosols from the sides and top of the masks.

This is an excellent point and we agreed that further research is needed to quantify the droplet dispersion in other directions.

Reply to the Reviewer 2's comments

11. Aerosol transmission is mostly accepted for COVID-19 now - the authors need to acknowledge this and update their literature review/Introduction/Discussion, to reflect this - but emphasise that they are focusing on the large droplet containment properties of various masks.

We thank the Reviewer for this important point. We addressed this in the first paragraph of the manuscript and in the second paragraph of the Discussion.

12. Also, I've reviewed this paper previously and found the video very unhelpful - the orientation is not very clear and the particles are not really visible. They need to improve this or just delete it.

The video is not included in the revised submission.

Face Coverings and Respiratory Tract Droplet Dispersion

Lucia Bandiera^{1*}, Geethanjali Pavar^{1*}, Gabriele Pisetta^{1*}, Shuji Otomo^{1*}, Enzo Mangano¹, Jonathan R. Seckl², Paul Digard^{3 #}, Emanuela Molinari⁴, Filippo Menolascina¹ and Ignazio Maria Viola^{1#}

Abstract

Respiratory droplets are the primary transmission route for SARS-CoV-2; a principle which drives social distancing guidelines. Evidence suggests that virus transmission can be reduced by face coverings, but robust evidence for how mask usage might affect safe distancing parameters is lacking. Accordingly, we set out to quantify the effects of face coverings on respiratory tract droplet deposition. We tested an anatomically-realistic manikin head which ejected fluorescent droplets of water, and human volunteers, in speaking and coughing conditions without a face covering, or with a surgical mask or a single layer cotton face covering. We quantified the number of droplets in flight using laser sheet illumination and UV-light for those that had landed at table height at up to 2 m. For human volunteers, expiratory droplets were caught on a microscope slide 5 cm from the mouth. Whether manikin or human, wearing a face covering decreased the number of projected droplets by > 1000-fold. We estimated that a person standing 2 m from someone coughing without a mask is exposed to over 10,000 times more respiratory droplets than from someone standing 0.5 m away wearing a basic single layer mask. Our results indicate that face coverings show consistent efficacy at blocking respiratory droplets and thus provide an opportunity to moderate social distancing policies. However, the methodologies we employed mostly detect larger (non-aerosol) sized droplets. If aerosol transmission is later determined to be a significant driver of infection, then our findings may overestimate the effectiveness of face coverings.

Keywords

COVID-19, face covering, surgical mask, handmade mask, social distancing, respiratory droplets.

¹ School of Engineering, ² Queen's Medical Research Institute, ³ The Roslin Institute, ⁴ School of Informatics, University of Edinburgh, Edinburgh, UK.

* These authors contributed equally to this work

Corresponding authors: Dr Ignazio Maria Viola, School of Engineering, University of Edinburgh, The King's Buildings, Edinburgh, EH9 3FB, UK, I.M.Viola@ed.ac.uk, Land: 0131 650 5622, Mob: 07776134181; Prof. Paul Digard, The Roslin Institute, University of Edinburgh, Easter Bush, Midlothian EH25 9RG, UK, Paul.Digard@roslin.ed.ac.uk, Land: 0131 651 9240, Mob: 07814128086.

1. Introduction

SARS-CoV-2 is primarily transmitted from virus-laden fluid droplets ejected from the mouth of an infected carrier. These droplets are either inhaled by a recipient, deposited on the recipient's mouth or conjunctiva, or deposited on a surface (thereby generating a fomite) and then mechanically transmitted through physical contact [1]. Droplet diameter varies according to the mode of exhalation (e.g. quiet breathing versus coughing) but ranges from ~100 nm to 1 mm [2], [3]. Intuitively, larger droplets potentially contain more virus, but the infectivity of viruses may vary between droplet size classes due to the influence of drying, making it extraordinarily difficult to determine the dominant mode of transmission [4]. Evidence indicates that transmission occurs both through the smallest droplets [5], [6], known as aerosol, as well as the direct deposition on the recipient's mouth, nose or conjunctiva of the largest droplets [2-4]. This paper focuses on the effect of face coverings in mitigating the dispersion of the largest droplets.

Recent reviews [5-7] suggest that face coverings are effective in decreasing the risk of infection. This has been inferred through epidemiological studies [13], tests with animals [14], and physical tests. For example, Viola et al. [15] showed that face coverings decreased the front throughflow of aerosol droplets by about an order of magnitude.

Comparatively, there is weaker evidence that face coverings mitigate dispersion of large respiratory droplets. Palacios et al. [16] sprayed bacterial-suspension droplets with a diameter from 20 μm to 900 μm through different household textiles and showed that this could mitigate large droplet dispersal. Yet, to allow an evidence-based assessment of what the appropriate social distance is when masks are worn, face fitted masks must be tested with realistic respiratory airflow jets and with human volunteers.

Recently, Anfinrud et al. [17] demonstrated that large respiratory droplets ejected by a person speaking can be visualised by laser sheet illumination. Successively Fischer et al. [18] used this technique to compare the effectiveness of different face masks in filtering respiratory droplets. A person spoke in front of a hole in an enclosed box, where droplets passing through a laser sheet were imaged and counted for different face coverings. These experiments demonstrated that face coverings can be effective in mitigating the dispersion of droplets ejected by a person speaking. On the other hand, these studies did not provide information on the distance travelled by the droplets, and thus on how face coverings can be used to review social distancing guidelines.

We adopted a similar technique to quantify the in-flight droplets ejected by an anatomically realistic manikin, with and without face covering, at several distances up to 2 m from the source. We considered both speaking and coughing conditions. Furthermore, we complemented these measurements with two other independent measuring techniques, UV light imaging and microscopy, which resulted in consistent results. Finally, we measured with microscopy the droplets ejected by six individuals and the results corroborated those of the manikin.

2. Methods

2.1 SPEECH AND COUGH SIMULATOR

The same simulator used by Viola et al. [15] was used. Air flow was generated using an air compressor capable of delivering up to 180 litre/min, while a 100 μM fluorescein solution in water for droplet generation was supplied by a TCS M400S micropump with a variable output

range of 0-2.7 litre/min. Liquid and air flows were connected to a purpose-built droplet generation system fitted inside an anatomically realistic, adult, medical simulation manikin's mouth (Resusci Anne QCPR) that creates controllable droplet sizes. Using shadow imaging (Supplementary Material), we estimated that the manikin ejected a continuous distribution of droplets ranging from less than $26 \pm 7 \mu\text{m}$, which is the size of the smallest droplet we could measure and that remained airborne, and up to $590 \pm 7 \mu\text{m}$. There is no agreement in the literature on the average droplet size distribution of humans while speaking and coughing [21], but several measurements suggest that the median is of the order of $10 \mu\text{m}$; and droplets of $500 \mu\text{m}$ occur at a frequency that is one or two orders of magnitude lower [22]. Hence the manikin's range includes the largest aerosol droplets and all droplets with ballistic trajectories ejected by humans.

Air/liquid flows were ejected from a 2 cm diameter circular "tracheal" opening with a velocity comparable to those of a person either speaking [19] (1 m s^{-1}) or coughing [20] (10 m s^{-1}). To ensure a significant number of landed particles could be detected prior to their evaporation, droplets were ejected from the manikin at a higher volume rate compared to a human being. Specifically, 20 min of speech test in the manikin corresponded to a subject counting from 1 to more than 10,000; 10 min of coughing tests modeled 60,000 human coughs.

Masks were either surgical or single layer woven cotton (see Supplementary Material). Examination of samples by phase contrast microscopy showed that the cotton masks had a close weave with gaps of around $50 \mu\text{m}$, while the multiple layers of less densely woven material in the surgical mask left a broader spectrum of pore sizes (Figure S1).

2.2 LASER IMAGING OF DROPLETS IN FLIGHT

We projected a thin laser sheet along the vertical plane through the mouth of the manikin and used a photographic camera to capture the light scattered by droplets passing within this plane. A 2.5 W diode-pumped continuous wave laser (532 nm) running at 40% of maximum power illuminated a plane perpendicular to the floor along the air flow jet axis. An 8-bit CCD camera with a resolution of 2056×2060 pixels with a Nikon 50 mm f/2 lens was used to image a physical plane of 137.5×137.8 mm, with a resolution of $67 \mu\text{m}/\text{pixel}$. A 60 ms exposure time was used to count particles under all conditions except coughing without a mask, where 30 ms was used. A total of 100 images from six replicate experiments were used for particle counting analysis. The light scattered by fluorescein made droplets appear larger than their actual size. Using a shadow imaging technique, we verified that the laser imaging visualised all droplets large enough to fall ballistically (see Supplementary Material).

2.3 UV-LIGHT IMAGING

Paper sheets were placed on a table 0.426 m below the manikin to cover areas of $0.84 \text{ m} \times 0.6 \text{ m}$ for speaking and $2.1 \text{ m} \times 0.3 \text{ m}$ for coughing. Speaking and coughing tests without a face covering lasted two and one minute, respectively, while durations of 20 and 10 minutes were used when the manikin was wearing a handmade or surgical mask. At least six replicates were run for each test (speaking, coughing) and condition (presence/absence of face covering). After each test, paper samples were placed on a Safe Imager 2.0 Blue-Light Transilluminator (Invitrogen™ G6600) to visualise deposited droplets. Images were acquired using an iPhone 7 camera (f/1.8, 4032×3024 pixels) with a resolution of $65 \mu\text{m}/\text{pixel}$. Images were imported into Inkscape to reconstruct the continuous sample and to equalise the pictures across the whole sample. Upon binarisation of the reconstructed image with a manually defined threshold, droplets were counted in each grid element using Fiji's edge-detection algorithm.

2.4 MICROSCOPY

Samples caught on glass microscope slides were imaged with a CFI Plan Fluor 20X objective mounted on a Nikon Eclipse Ti inverted microscope, equipped with an iXon Ultra 888 EMCCD camera, resulting in a resolution upon magnification of $0.65 \mu\text{m}/\text{pixel}$. For each sample, 210 fields of view, covering an area of 0.85 cm^2 in total, were acquired in the FITC channel (Gain 70, Exposure time 100 ms) using NIS-Elements. Samples collected in tests involving human subjects were imaged with an Andor Zyla sCMOS camera and the same objective and microscope as above, resulting in a resolution upon magnification of $0.33 \mu\text{m}/\text{pixel}$. We acquired 153 fields of view, corresponding to 0.62 cm^2 . All images were downscaled by a factor of five prior to their analysis in scikit-image, Python. Droplets in each sample were counted as the markers of the watershed algorithm, which allows segmentation of touching objects in an image. A Gaussian smoothing filter was applied (standard deviation of five) to reduce pixel noise before a thresholding procedure using a manually selected value. Following morphological opening with a Boolean kernel of size seven, droplet centres were identified as the local maxima of the distance transform of the image, with the stipulation that centres should be more than 15 pixels from each other.

2.5 TESTS WITH HUMAN VOLUNTEERS

Six volunteers performed two rounds of coughing (1 min/each) and speaking (3 min/each) tasks, with and without a surgical mask. Speech tasks were performed reading a provided sample text, to ensure results were not biased by the personal choice of words. On the two days of the trial experiment, the order in which the tasks were performed was reversed. This research protocol was approved by the University of Edinburgh's Human Research Ethical Review Committee and all human participants gave written informed consent.

2.6 STATISTICAL ANALYSIS

The mean and the standard error of the mean (SEM) of the data are presented. Due to inhomogeneity of variance, results were assessed by Bonferroni-corrected Kolmogorov-Smirnov statistical tests applied to pairwise combinations of no-mask/handmade mask, no-mask/surgical mask, and handmade mask/surgical mask. Individual statistical tests were performed for each distance from the source.

3. Results

3.1 IMAGING DROPLETS IN FLIGHT

We recorded images at eight different positions, whose vertical and horizontal distances are presented in Fig. 1A. Fluorescent droplets appeared as segments on the images, with a length proportional to their speed (Figure S2). We observed three types of droplets: droplets smaller than $\sim 30 \mu\text{m}$ that remained airborne following air currents in the room, droplets larger than $\sim 170 \mu\text{m}$ that fell ballistically with trajectories similar to the red lines in Figure 1A, and intermediate size droplets which could show any of the two behaviours or fall with a non-ballistic trajectory. The size of the droplets was assessed using shadow imaging technique (Supplementary Material).

Visually, the impact of placing a mask on the manikin was obvious, especially under coughing conditions (Figure 1B). To quantify the data, we counted the number of particles crossing the lower edge of the field of view at positions 1 to 7, therefore extrapolated to have deposited on

the table. Data were compiled from 100 images taken from six replicate runs, each with no face covering, with a surgical mask or with a single-layer face covering. We used a new mask for each repeat to account for potential variability in the masks. The absolute particle count was divided by the length of the lower edge of the field of view and by the total period of observation to provide the rate of deposition of particles along the centreline of the table. Tests modelling speaking without a mask showed an exponential decline in particle deposition rate with distance, with none reaching the 2 m mark (Fig. 1C), while the distribution peaked at 1 m and dropped more slowly for coughing (Fig. 1D). When the manikin was fitted with either surgical or handmade masks, not one droplet was found to fall through the lower edge of the fields of view in either speaking or coughing simulations. Therefore, both face masks were largely impermeable to droplets larger than 20-30 μm .

3.2 DROPLET DEPOSITION

To provide an overview of the spread of droplets over the table, we first used UV light to image the distribution of fluorescent particles that accumulated on white paper placed in front of the manikin. An example result for a speaking test is shown in Figure S3. We measured droplet deposition rate according to distance, presented as the number of particles divided by the sampling area and the duration of the experiment by counting the droplets every 5 and 15 cm from the manikin for speaking and coughing, respectively. Each experiment was performed six times with and without both types of face-covering, using a new mask for each repeat. The numbers of surface-deposited droplets were consistent with the measurements of falling droplets (Figure 2A, B). Once again, the presence of a face-covering very effectively blocked droplet deposition. However, because we measured droplets on a larger area and longer duration (up to 20 min) than the flight tests, we could capture less frequent events. With both mask types, small numbers of droplets (up to seven) were seen at various distances from the manikin, but these values were four and five orders of magnitude lower than those observed without a mask for speaking and coughing, respectively. Hence, these represent events with a frequency of one in 10,000 and one in 100,000, respectively.

To further corroborate our results, we imaged deposited droplets using microscopy. To this end, the manikin was set to emulate speaking or coughing as before and fluid droplets were collected on glass slides positioned at varying distances (0.25, 0.5 and 0.75 m for speaking conditions and 0.5, 1, 1.5 and 2 m for coughing conditions) from the manikin. We repeated each experiment six times each with and without masks, using a new mask for each test. After each run the glass slides were collected and imaged with a 20X objective to count the number of droplets deposited in both cases. We then reconstructed the spatial distribution of droplets with and without masks (Figure 2 C, D). Consistent with other means of measurement, both types of face covering were extremely effective at reducing the particle deposition rate. As with UV imaging of droplets deposited on white paper, this technique also allowed a long experimental duration (20 min) and thus the capture of statistically rare events. With both mask types, we observed a few (up to 4) droplets at various distances from the manikin, including at 2 m for coughing. However, this number was again 4 - 5 orders of magnitude lower than the number of droplets observed without a mask for speaking and coughing, respectively.

3.3 DROPLET DEPOSITION FROM HUMAN VOLUNTEERS

Results obtained from a manikin, one might argue, are only indicative due to the wide variability in the expiration parameters for speech and coughing in humans [16,17]. Accordingly, we performed experiments using four male and two female volunteers (aged between 30 and 45) to either read a sample script for three minutes or cough for one minute. For each test, a glass slide was placed vertically, 5 cm in front of the individual's mouth. At the

end of each experiment, the sample was imaged using widefield microscopy to count the droplets generated by the subject. Each speaking and coughing test was performed twice each with and without a surgical mask on separate days, reversing the order of the tests on day two. In contrast to the manikin tests, there was a large variability in the number of droplets emitted by the individual subjects in the absence of a mask (Figure 3). However, for all subjects, we did not find a single droplet when a mask was worn.

Figure 1. Laser Imaging of Respiratory Droplets in Flight. (A) Schematic diagram of the experimental setup. Boxes indicate imaging windows. (B) Examples of images captured at position A (directly in

front of the mouth) for speaking (upper row) and coughing (lower row), without mask (1st column), with the surgical mask (2nd column) and with the handmade mask (3rd column). (C, D) Droplet deposition rate over the table centreline versus the horizontal distance from the manikin's mouth in (C) speaking and (D) coughing conditions. Data were estimated from the count of imaged droplets crossing the lower edge of the field of view per unit time and are reported as the mean ± 1 SEM of six independent replicates. (C) Both mask types statistically significantly reduced the droplet deposition rate at 0.21 m from the mouth of the source ($p = 0.016$ in both cases). Past 0.21 m, too few droplets were detected also without mask to observe statistically significant differences among the tested conditions. (D) No significant differences were detected between mask types, but both mask types statistically significantly reduced the droplet deposition rate at every tested position ($p = 0.019$).

Figure 2. Surface Deposition of Droplets. Droplet deposition rate versus horizontal distance from the manikin's mouth in (A, C) speaking or (B, D) coughing conditions, measured by (A, B) UV illumination of paper sheets or (C, D) microscopic imaging of glass slides placed on the table centre line. Data are the mean ± 1 SEM of six independent repeats. Statistical analyses indicated that for (A), both mask types significantly reduced the droplet deposition rate at up to 47.5 cm from the source ($p = 0.047$). For (B), droplet deposition rate with and without masks were not significantly different ($p = 0.055$). For (C), both mask types significantly reduced droplet deposition rate up to 0.5 m from the source ($p = 0.012$). Past that point, droplets were undetectable in all cases. For (D), both mask types significantly reduced droplet deposition rate at all distances from the source ($p = 0.016$). No statistically significant difference was observed between the two mask types in any test.

Figure 3. Droplet Deposition from Human Volunteers. Droplet deposition rate in human volunteers for the no-mask and surgical mask cases under (A) speech and (B) coughing conditions. Data represent the number of droplets imaged in widefield microscopy on a glass slides placed vertically at a distance of 5 cm from the human volunteer performing the task. To estimate the deposition rate, droplet counts were divided by the imaged area and duration of the speech or coughing task. Measurements are plotted as a box between the 25th and 75th percentiles with a mark indicating the median. The whiskers extend over the full data range. Statistical tests showed $p = 2.3 \times 10^{-6}$ for both cases, thus rejecting the null hypothesis that the two samples were drawn from the same distribution. Therefore, the use of surgical masks statistically significantly reduced droplet deposition rate.

4. Discussion

We employed three independent, quantitative techniques to measure the droplet deposition rate at increasing distance from a manikin. Without any face covering, we measured distributions of deposition rates comparable to those reported in the literature [18-20]. When the manikin wore any of the two face masks, we observed that less than one in 1,000 particles escaped for both speaking and coughing. To further demonstrate that our results could be applied to real people speaking and coughing, we tested six individuals using microscopy. Without masks, we measured between 10 s and 1000 s of particles for speaking and coughing without a mask, respectively. Conversely, we found zero particles with the surgical mask, both for speaking and coughing.

Overall, these data demonstrate that face masks are highly effective at reducing exhalation of large respiratory droplets. As these droplets are likely to be the main driver of SARS-CoV-2 transmission [2,3,8], our data suggest that the wearing of masks can substantially reduce the probability of an infected person transmitting the virus. In this study, however, we focused only on the large respiratory droplets that land on a surface within few seconds. According to the shadow imaging estimation of droplet size applied to the laser acquisition (Supplementary Material, S4), these correspond to particles with diameters between $170 \pm 7 \mu\text{m}$ and $590 \pm 7 \mu\text{m}$. Droplets as small as $26 \pm 7 \mu\text{m}$ were detected with the laser but excluded from the analysis due their non-ballistic trajectory. Other studies such as, for instance Fischer et al. [18], which focuses on in-flight droplets and that include smaller droplets and aerosol, are likely to show a lower level of effectiveness depending on the fabric type. Aerosol is now widely accepted as a contributory route of SARS-CoV-2 virus transmission [5], and if this is later determined to be the main driver of infection, then our findings may overestimate the effectiveness of face coverings.

Air was ejected at velocities and flow rates within the range of those observed from real individuals [14,15] but droplets were ejected at a higher volume rate. Hence, the physical interaction between the droplets and the airflow jet [16,20] was not correctly represented.

However, this was necessary to ensure a high volume of particles and thus robust statistics, in a sufficiently short time frame to minimise evaporation of landed droplets and contamination with other particles in the environment. By counting from 1 to 100, a person ejects between 10 and 100 mg of water [25], whilst our manikin ejected about 10 mg s⁻¹. Thus, each second of test corresponded to a person counting at least from 1 to 100. The duration of the tests was as high as 20 minutes; hence we modelled a person counting up from 1 to more than 10,000. In one cough a person ejects about 1 mg of fluid [25], whilst our manikin ejected 100 mg s⁻¹. We tested cough for as long as 10 minutes, which is equivalent to 60,000 coughs. Furthermore, it was not possible to test if droplets escaped the mask and landed outside of the table. Viola et al. [15] showed that surgical and handmade masks can lead to lateral and backwards air jets. If a large droplet was carried by these jets, we would have not detected it. Finally, we tested simple single layer handmade masks, but it should be recognised that there is a wide range of handmade masks and some might not be fit for purpose.

In conclusion, these experiments demonstrate that both surgical and simple handmade masks such as a single layer cotton mask are can suppress the risk of direct person-to-person virus transmission through large droplet deposition. The data do not allow us to draw conclusions on the risks of virus transmission through aerosol inhalation. Assuming that SARS-CoV-2 virus transmission through aerosol is small compared to through large droplets, these results suggest that physical distancing can be reduced with the use of face coverings.

Declaration of interests

None of the authors has any conflict of interest.

Acknowledgements

The authors are grateful to Dr B. Peterson (School of Engineering) and Dr A. Nila (LaVision) for their advice on the laser tests and for lending related equipment, and to Dr K. Dunn (School of Engineering), Prof. F. Denison (Queen's Medical Research Institute) and Dr F. Mehendale (Usher Institute) for procuring the tested face masks.

Bandiera is supported by the UK Engineering and Physical Sciences Research Council (EPSRC), grant no. EP/P017134/1. Pavar and Pisetta are supported by the EPSRC grant EP/L016680/1, while Molinari by the EPSRC grant EP/S02431X/1. Ōtomo's scholarship is funded by the Japan Student Services Organization. Menolascina is supported by the European Commission (766840) and the EPSRC (EP/S001921/1 and EP/R035350/1), whilst Digard is funded by the UK Biotechnology and Biological Sciences Research Council (BB/P013740/1).

Author Contributions

Bandiera, Menolascina, Otomo, Pavar and Pisetta undertook the experiments. Mangano led the design and manufacturing of the speech and cough simulator. Menolascina conceived and led the microscopy investigation. Digard, Menolascina, Molinari, Seckl and Viola designed the experiments. Viola coordinated the project. Molinari wrote the first draft of this document that was edited, reviewed and approved by all the authors.

References

- [1] R. M. Jones and L. M. Brosseau, “Aerosol Transmission of Infectious Disease,” *J. Occup. Environ. Med.*, vol. 57, no. 5, pp. 501–508, May 2015.
- [2] S. Asadi, A. S. Wexler, C. D. Cappa, S. Barreda, N. M. Bouvier, and W. D. Ristenpart, “Aerosol emission and superemission during human speech increase with voice loudness,” *Sci. Rep.*, vol. 9, no. 1, p. 2348, Dec. 2019.
- [3] R. S. Papineni and F. S. Rosenthal, “The Size Distribution of Droplets in the Exhaled Breath of Healthy Human Subjects,” *J. Aerosol Med.*, vol. 10, no. 2, pp. 105–116, Jan. 1997.
- [4] R. Tellier, Y. Li, B. J. Cowling, and J. W. Tang, “Recognition of aerosol transmission of infectious agents: a commentary,” *BMC Infect. Dis.*, vol. 19, no. 1, p. 101, 2019.
- [5] L. Morawska and D. K. Milton, “It is Time to Address Airborne Transmission of COVID-19,” *Clin. Infect. Dis.*, pp. 1–23, 2020.
- [6] R. Zhang, Y. Li, A. L. Zhang, Y. Wang, and M. J. Molina, “Identifying airborne transmission as the dominant route for the spread of COVID-19,” *PNAS*, 2020.
- [7] SAGE – Environmental and Modelling Group 4th June, “Transmission of SARS-CoV-2 and Mitigating Measures,” 2020.
- [8] R. Dhand and J. Li, “Coughs and Sneezes: Their Role in Transmission of Respiratory Viral Infections, Including SARS-CoV-2.,” *Am. J. Respir. Crit. Care Med.*, pp. 1–37, 2020.
- [9] S. Romano-Bertrand, L.-S. Aho-Glele, B. Grandbastien, J.-F. Gehanno, and D. Lepelletier, “Sustainability of SARS-CoV-2 in aerosols: Should we worry about airborne transmission?,” *J. Hosp. Infect.*, 2020.
- [10] J. S. Brainard, N. Jones, I. Lake, L. Hooper, and P. Hunter, “Facemasks and similar barriers to prevent respiratory illness such as COVID-19: A rapid systematic review,” *medRxiv*, p. 2020.04.01.20049528, 2020.
- [11] J. Howard *et al.*, “Face Masks Against COVID-19: An Evidence Review,” *Preprints*, no. 2020040203, pp. 1–8, 2020.
- [12] D. K. Chu *et al.*, “Physical distancing, face masks, and eye protection to prevent person-to-person transmission of SARS-CoV-2 and COVID-19: a systematic review and meta-analysis.,” *Lancet*, vol. 6736, no. 20, 2020.
- [13] N. H. L. Leung *et al.*, “Respiratory virus shedding in exhaled breath and efficacy of face masks,” *Nat. Med.* 2020, pp. 1–5, Apr. 2020.
- [14] J. F.-W. Chan *et al.*, “Surgical mask partition reduces the risk of non-contact transmission in a golden Syrian hamster model for Coronavirus Disease 2019 (COVID-19),” *Clin. Infect. Dis.*, May 2020.
- [15] I. M. Viola *et al.*, “Face Coverings, Aerosol Dispersion and Mitigation of Virus Transmission Risk,” *Prepr. arXiv2005.10720*, 2020.
- [16] A. Rodriguez-Palacios, F. Cominelli, A. R. Basson, T. T. Pizarro, and S. Ilic, “Textile Masks and Surface Covers—A Spray Simulation Method and a ‘Universal Droplet

- Reduction Model' Against Respiratory Pandemics," *Front. Med.*, vol. 7, no. May, pp. 1–11, 2020.
- [17] P. Anfinrud, V. Stadnytskyi, C. E. Bax, and A. Bax, "Visualizing Speech-Generated Oral Fluid Droplets with Laser Light Scattering," *N. Engl. J. Med.*, vol. 382, no. 21, pp. 2061–2062, 2020.
- [18] E. P. Fischer, M. C. Fischer, D. Grass, I. Henrion, W. Eric, and S. W. Warren, "Low-cost measurement of facemask efficacy for filtering expelled droplets during speech," *ScienceAdvances*, p. 11, 2020.
- [19] J. W. Tang *et al.*, "Airflow Dynamics of Human Jets: Sneezing and Breathing - Potential Sources of Infectious Aerosols," *PLoS One*, vol. 8, no. 4, p. e59970, Apr. 2013.
- [20] Z. Y. Han, W. G. Weng, and Q. Y. Huang, "Characterizations of particle size distribution of the droplets exhaled by sneeze," *J. R. Soc. Interface*, vol. 10, no. 88, p. 20130560, Nov. 2013.
- [21] G. Seminara, B. Carli, G. Forni, S. Fuzzi, A. Mazzino, and A. Rinaldo, "Biological fluid dynamics of airborne COVID-19 infection," *Rend. Lincei*, vol. 31, no. 3, pp. 505–537, 2020.
- [22] C. Y. H. Chao *et al.*, "Characterization of expiration air jets and droplet size distributions immediately at the mouth opening," *J. Aerosol Sci.*, vol. 40, no. 2, pp. 122–133, Feb. 2009.
- [23] R. Mittal, R. Ni, and J.-H. Seo, "The Flow Physics of COVID-19," *J. Fluid Mech.*, pp. 1–14, 2020.
- [24] X. Xie, Y. Li, A. T. Y. Chwang, P. L. Ho, and W. H. Seto, "How far droplets can move in indoor environments - Revisiting the Wells evaporation-falling curve," *Indoor Air*, vol. 17, no. 3, pp. 211–225, Jun. 2007.
- [25] X. Xie, Y. Li, H. Sun, and L. Liu, "Exhaled droplets due to talking and coughing," *J. R. Soc. Interface*, vol. 6, no. SUPPL. 6, Dec. 2009.
- [26] L. Bourouiba, E. Dehandschoewercker, and J. W. M. M. Bush, "Violent expiratory events: On coughing and sneezing," *J. Fluid Mech.*, vol. 745, pp. 537–563, Apr. 2014.